# Characteristics of *Fragaria vesca* Yield Parameters and Anthocyanin Accumulation under Water Deficit Stress

**DOI:** 10.3390/plants10030557

**Published:** 2021-03-16

**Authors:** Rytis Rugienius, Vidmantas Bendokas, Tadeusas Siksnianas, Vidmantas Stanys, Audrius Sasnauskas, Vaiva Kazanaviciute

**Affiliations:** 1Lithuanian Research Centre for Agriculture and Forestry, Department of Orchard Plant Genetics and Biotechnology, Institute of Horticulture, LT-54333 Babtai, Lithuania; vidmantas.bendokas@lammc.lt (V.B.); jurate.siksnianiene@lammc.lt (T.S.); vidmantas.stanys@lammc.lt (V.S.); audrius.sasnauskas@lammc.lt (A.S.); 2Department of Eukaryote Gene Engineering, Institute of Biotechnology, Vilnius University, LT-10257 Vilnius, Lithuania; kaaiva@yahoo.de

**Keywords:** anthocyanins, deficit irrigation, drought, pigments, temperature

## Abstract

Plants exposed to drought stress conditions often increase the synthesis of anthocyanins—natural plant pigments and antioxidants. However, water deficit (WD) often causes significant yield loss. The aim of our study was to evaluate the productivity as well as the anthocyanin content and composition of berries from cultivated *Fragaria vesca* “Rojan” and hybrid No. 17 plants (seedlings) grown under WD. The plants were grown in an unheated greenhouse and fully irrigated (control) or irrigated at 50% and 25%. The number of berries per plant and the berry weight were evaluated every 4 days. The anthocyanin content and composition of berries were evaluated with the same periodicity using HPLC. The effect of WD on the yield parameters of two evaluated *F. vesca* genotypes differed depending on the harvest time. The cumulative yield of plants under WD was not less than that of the control plants for 20–24 days after the start of the experiment. Additionally, berries accumulated 36–56% (1.5–2.3 times, depending on the harvest time) more anthocyanins compared with fully irrigated plants. Our data show that slight or moderate WD at a stable air temperature of about 20 °C positively affected the biosynthesis of anthocyanins and the yield of *F. vesca* berries.

## 1. Introduction

Abiotic stresses, including drought, have detrimental effects on the yield of various crops in different parts of the world, and agriculture in many regions has become uncompetitive as a result of this yield instability. Predicted climate changes in the near future are expected to cause major challenges in modern agriculture due to their significant potential negative impacts on both the quantity and quality of various crops, including strawberry [1,2]. Water deficit is generally associated with a reduction in the size and yield of strawberry fruits. However, some studies have demonstrated that water deficit might increase the dry matter content, improve the concentration of health-related compounds, and improve the taste of strawberry fruits [3,4,5,6,7]. Declining water resources are leading to concerns about high water usage for some horticultural purposes. Studies to determine the minimum necessary amount of water for crop development are required not only to save water and/or energy, but also to improve plant fitness to resist biotic and abiotic stresses, as well as improve nutritional, functional, and sensorial food properties [5,8].

Plants exposed to stress often increase the biosynthesis of phenolic compounds, particularly flavonoids, such as anthocyanins. As one of the most ubiquitous classes of flavonoids in berries, anthocyanins possess a multitude of biological roles, including protection against solar exposure, and ultraviolet radiation, free radical scavenging and anti-oxidative capacity, defense against many different pathogens, attraction of predators for seed dispersal, and the modulation of signaling cascades [9]. Epidemiological and clinical studies suggest that an anthocyanin-enriched diet may lower levels of certain oxidative stress biomarkers in humans, and this could be associated with a reduced risk of cognitive decline and the development of neurodegenerative and cardiovascular diseases, in addition to sustaining hepatic function and kidney protecting activities [10]. Berry anthocyanins improve neuronal and cognitive brain functions and ocular health as well as protect genomic DNA integrity. Anthocyanins assist in antiplatelet aggregation, act as an antiangiogenic barrier, and display anticancer properties [11,12].

Anthocyanin composition, as well as that of other flavonoids, has been studied extensively in wild and cultivated strawberry [13,14,15,16,17,18].This notwithstanding, the relation between water deficit (WD) and anthocyanin accumulation is not fully understood, especially in fruit crops, and the potential to increase food value and reduce cultivation costs is not fully exploited. Wild strawberry (*Fragaria vesca*) is a good model plant for various studies because of its short cycle, small genome, and ease of reproduction [19]. Wild strawberry is a relative of the cultivated strawberry *Fragaria × ananassa*, which is grown in open fields and greenhouses. The aim of our study was to evaluate the growth and fruiting of *F. vesca* plants and their anthocyanin content under water deficit in a greenhouse. 

## 2. Results

### 2.1. F. vesca Yield Parameters

The number and yield of berries from fully irrigated control (I100) plants of both “Rojan” and hybrid No. 17 remained relatively stable during the first two weeks of the treatment in April (Figure 1a,b,e,f). Later, starting from the fourth–fifth pick, or 20–25 days after the beginning of harvesting, the number and yield of berries started to increase. Berry ripening dynamics were affected by extreme air temperature, which increased to 24 °C outside starting on 8 May 2013. The temperature inside the greenhouse sometimes reached 35 °C. The rise in temperature accelerated berry ripening and led to increases in the number and yield of ripened berries. This increase was variable and reached a maximum on 20–24 May. The berry size decreased gradually until 8 May. Then, a temporary increase in berry size, especially for hybrid No. 17, was observed (Figure 1c,d). This was also clearly a consequence of the increased air temperature. 

The impact of water deficit on yield parameters was genotype dependent. This impact was much more noticeable in hybrid No. 17 than in “Rojan”. The cumulative berry number of hybrid No. 17 plants grown under water deficit was higher than that of fully irrigated plants for almost the entire treatment period (Figure 1b). For 8 days of the experiment, the cumulative yield of 50% irrigated plants (I50) did not differ from that of control (fully irrigated) plants, and their yield subsequently became significantly higher and was the highest at the end of the experiment compared with 25% irrigated (I25) and control plants (Figure 1f). The cumulative yield of I25 plants was higher than that of I100 plants until 16 May, which is about two-thirds of the duration of the treatment. During the last 4 days, it became the lowest compared with I50 and control plants. Interestingly, although water deficit significantly (except for 8 May) decreased the average weight of the fruit of this hybrid, for most of the harvesting period, the cumulative yield was higher than that of I100 plants (Figure 1d). 

The cumulative berry number of “Rojan” plants grown under water deficit was also higher than that of control plants for almost the entire duration of the experiment. However, the difference was much smaller than that of No. 17 plants, and at the end of the experiment, the number of berries was the same among all irrigation variants of “Rojan” (Figure 1a). The cumulative yield of I50 “Rojan” plants did not differ from that of I100 plants for the entire experimental period. The yield of I25 plants from the 8th to 24th day of the experiment (30 April–20 May) was higher than that of the control, but starting on 24 May, it decreased, and at the end of the experiment, it was the lowest compared with other irrigation variants (Figure 1e). The difference in average berry weight among different treatments was also much lower in “Rojan” than in hybrid No. 17 (Figure 1c). The berry weight of I50 plants did not differ from that of I100 plants for almost the entire experimental period. Starting on 16 May, the berry weight of I25 plants significantly decreased compared with that of the control, and it was the lowest at the end of the experiment. This decrease, possibly combined with the lack of an increase in the number of berries, resulted in I25 having the lowest yield compared with the control and I50 plants at the end of experiment. 

In hybrid No. 17 plants under half irrigation (I50), the total yield increase was 6.2% compared with control plants. The decreases in total yield per plant of hybrid No. 17 and “Rojan” I25 plants were 8.6% and 21.0%, respectively. 

The experiment for the least irrigated (I25) plants was completed on 24 May, when plants became dry, and they died before 28 May. Clear signs of wilting were visible in the middle of May.

### 2.2. Anthocyanin Profile and Concentration in Berries during Harvesting

The anthocyanin concentration of fully irrigated *F. vesca* berries of both varieties varied from 3.6 to 4.7 mg/100 g fresh weight and remained stable until 4 May (Figure 2a,c). Then, the anthocyanin concentration decreased more than two-fold in both varieties until the end of the harvesting season. It was observed in our study that WD caused a considerable increase in anthocyanin content in wild strawberry fruits, especially at the beginning of the harvest. The berries of “Rojan” I50 and I25 plants accumulated 1.5 and 2.3 times more anthocyanins then fully irrigated plants after 4 days of the treatment (Figure 2a). Anthocyanin concentrations in the berries of hybrid No. 17 were 1.5–1.6 times higher than those of fully irrigated plants under both deficit irrigation regimes (Figure 2c). Differences in the total amount of berry anthocyanins obtained under full and deficit irrigation regimes were significant (*p* ≤ 0.05) for both cultivars. When analyzing the cumulative anthocyanin content at the end of study, we observed clear differences between irrigation treatments starting on the eighth day of the treatment for “Rojan” and on the 12th day for hybrid No. 17. At the end of the study, the cumulative amount of anthocyanins in berries of I25 both “Rojan” and No. 17 plants was 56% and 36% higher, respectively, than that of fully irrigated control plants (Figure 2b,d). 

We analyzed the anthocyanin composition of *F. vesca* berries and established that pelargonidin 3-O-glucoside (Pel3G) was the most abundant anthocyanin. In “Rojan” and No. 17 berries, Pel3G accounted for 50.7% ± 7.5% and 50.6% ± 4.4% of all anthocyanins, respectively, followed by cyanidin-3-O-glucoside (C3G), which accounted for 29.2% ± 8.3% and 27.4% ± 5.5% of all anthocyanins, respectively. The percentage of peonidin-3-O-glucoside was 8.4% ± 1.2% and 10.1% ± 1.7%, respectively. The concentrations of other unidentified minor anthocyanins were low and together accounted for less than 10% of total anthocyanins. In examining the ratio of anthocyanins, we focused on the first two anthocyanins, as their levels were the highest. A tendency of C3G to decrease and Pel3G to increase during harvesting was clearly apparent, although the variation was slight throughout the experiment. This tendency was more consistent in the berries of “Rojan”. The Pel3G/C3G ratio in the berries of this cultivar was 1.31 at the start of the treatment and was relatively stable (up to 12 days after the start of treatment), and then it increased significantly: on the 28th day, it was 2.9 times higher compared with the beginning of the study period (Figure 3a). In contrast to this variety, a different trend was established in berries of hybrid No. 17: the Pel3G/C3G ratio decreased at the beginning and subsequently began to increase, and it had increased by 1.4 times at the end of the study compared with the start (Figure 3b). The proportion and dynamics of both main anthocyanins in berries during harvesting did not depend on water deficit. However, the Pel3G/C3G ratio in the berries of No. 17 plants irrigated at 25% (I25) was up to 35% and 20% lower in the beginning and at the end of the study compared with that in fully irrigated plants.

## 3. Discussion

Water deficit in strawberry fruits is generally associated with a reduction in berry size and yield. Previous studies have shown that drought stress decreases yield and fruit weight, as well as leaf area, leaf dry matter, shoot dry matter, total dry matter, relative water content, and stomatal conductance [3,5,20,21]. Liu and co-authors [22] showed that water deficit and partial root-zone drying decreased the berry weight of the strawberry (“Honeoye”), but the berry number remained stable.

The impact of our WD treatments on different yield parameters (berry number and berry weight) of wild strawberry depended on the variety. Insufficient irrigation resulted in decreased berry weights for strawberry hybrid No. 17 and, to a much lesser extent, “Rojan”, but it increased the berry number and yield at the beginning of the treatment. WD stress has been reported to stimulate both berry set and ripening speed [23]. Mild WD was found to increase apple yield [24]. An increase in berry yield per plant of a strawberry hybrid was observed in our other experiment [25]. This indicates that the dehydration stress-induced increase in strawberry yield, at least during certain growth stages, is not coincidental. 

Differences in berry set in response to water stress between the evaluated varieties may be explained by the genetic background of hybrid No. 17. One of its parents is the woodland strawberry *F. vesca*, which has probably evolved with different adaptive characteristics from those of the cultivated alpine strawberry. 

Differences among genotypes have been observed by other authors investigating the response of strawberry plants to WD [20,21,22,23,24,25,26,27]. The reaction to water deficit stress in plants of different genotypes has been related to the ratio between carbon fixation and water loss [20] or differences in the response to jasmonic acid and stomatal closure speed [28].

The rise in temperature before 8 May further accelerated the ripening of berries in all studied irrigation regimes. An increase in berry yield due to elevated air temperature was also observed in our other experiment [25]. In both cases, berry weight and yield decreased at the end of harvest, especially when plants were grown under WD. 

In our opinion, the changes in yield parameters under water deficit conditions in our treatment can be divided into three stages. In the first stage, the air temperature was low and stable, and water deficit caused berry ripening to slightly accelerate. The decrease in the weight of the berries was variable and depended on the variety. In the second stage, when the air temperature rose and fluctuated, the ripening of the berries accelerated even further, but the stress due to water deficit also increased. The berry size decreased more rapidly under a larger water deficit. In the third stage, the negative effects of water deficit stress, exacerbated by higher temperatures, revealed themselves. The size of the berries decreased substantially, the number of berries did not increase, and the yield decreased.

Plants need to endure different abiotic stresses, and polyphenols (including anthocyanins) accumulate in response to these stresses, helping plants to acclimatize to unfavorable environments. Hence, the concentration of these metabolites in plant tissue is a good indicator to predict the extent of abiotic stress tolerance in plants, which varies greatly among plant species under an array of external factors [29].

As mentioned before, in our experiment, water deficit caused a considerable increase in anthocyanin content in wild strawberry fruits, especially at the beginning of the harvest. According to data from several studies, water stress induces the accumulation of anthocyanin compounds in the berries of strawberry plants [5,21,30]. Bordonaba and Terry [3] showed that differences exist in the way that different strawberry cultivars respond to drought stress, resulting in different fruit compositions. Despite the detrimental effect that WD can have on berry size, reducing water irrigation by a quarter between flower initiation and fruit harvest has been found to result in better water use efficiency, as well as enhanced fruit quality and taste in some cultivars [3]. Mild drought and salt stress resulted in an increased content of phenolics, anthocyanins, and l-ascorbic acid in strawberry fruits; however, fruit yield was not affected [5].

According to the data from our study, increase in anthocyanin content in wild strawberry fruits under water deficit had two peaks: the first was at the beginning of the treatment, and the second was after the middle of the treatment, from 8 May. We propose that the first peak was caused by slight or moderate water deficit stress, and the second peak was the result of air temperature changes, and an increase in sunny hours. It is known that the accumulation of anthocyanins and other polyphenols in plants is influenced by many factors, including light and temperature [29]. Interestingly, anthocyanin accumulation had two peaks in our other experiment [25]. The influence of temperature on anthocyanin biosynthesis is ambiguous according to different investigators. Higher temperature during cultivation generally promotes the synthesis of anthocyanins and other phenolic compounds in strawberry fruits [31], but this effect is genotype dependent [1]. According to Wang and Camp [32], a daytime temperature of 30 °C inhibited plant and fruit growth and also reduced fruit quality. However, elevated CO_2_ and a higher temperature enhanced the amounts of accessible bioactive compounds in strawberries, including anthocyanins [1]. Generally, temperatures of approximately 25 °C favor anthocyanin biosynthesis, whereas higher temperatures, such as 35 °C, are associated with anthocyanin degradation [33]. In our treatment, anthocyanin content in the berries of “Rojan” plants grown under water deficit increased at the beginning of harvesting until a sharp temperature increase occurred. However, the increase in temperature and sunlight may have affected the fluctuations in anthocyanin levels in the second half of harvesting. The results of our study show that WD has a positive effect on anthocyanin accumulation in *Fragaria* berries only at moderate air temperatures of around 20 °C. Similar results were obtained in our other study, in which anthocyanin accumulation did not increase under WD at elevated temperatures [25]. In the current experiment, we observed a substantial decrease in the anthocyanin content in berries starting on 13 May, when the air temperature inside the greenhouse increased to over 30 °C. It is probable that anthocyanin synthesis and accumulation reached a certain limit at higher air temperatures, as observed in other studies [34,35]. The anthocyanin content in the berries of hybrid No. 17 decreased 8 days later, on 21 May. This shows that the accumulation of anthocyanins in the berries of hybrid No. 17 is less sensitive to temperature despite the higher amplitude of the change in morphological yield parameters; that is, an increase in berry number and decrease in average berry weight, compared with the “Rojan” variety. These results corroborate the findings of Crespo et al. [36] and Carbone et al. [15], who observed that the strawberry genotype had a dominant role over environmental factors in anthocyanin accumulation. Anthocyanins, phenolic acids, sucrose, and malic acid were found to be the most discriminant variables among cultivars, while climatic conditions and the cultivation system were behind changes in polyphenol contents [16].

As mentioned before, pelargonidin 3-O-glucoside (Pel3G) and cyanidin-3-O-glucoside (C3G) in *F. vesca* berries in our treatment constituted nearly 50.7% and 27.4–29.2% of total anthocyanins, respectively. According to the data of Kawanobu et al. [37], in the berries of wild strawberry, C3G and Pel3G constituted 49.2% and 40.7% of all anthocyanins, respectively. Pel3G clearly predominates, reaching from 50% to more than 90% of total anthocyanin content, while other anthocyanins (Pel3Rut, C3G, and Pel3MG) rarely exceed 30% in *F. ananassa*, according to data from different studies [13,37,38]. Lower Pel/C3G ratios are thought to be desirable because cyanidin 3-glucoside, in most cases, has greater antioxidant capacity than that of pelargonidin 3-glucoside [39]. 

Although the content of individual anthocyanins was influenced significantly by the strawberry cultivar, production system, harvest period, climatic conditions, soil, day length, and altitude, the proportions among Pel3G, C3G, and Pel3rut were consistent, suggesting a characteristic anthocyanin distribution [40,41]. In our experiment, an increase in the Pel3G/C3G ratio was observed in the second half of the growing season. This may be related to an increase in air temperature, which started in the middle of the harvest season. The percentages of separate anthocyanins did not differ significantly under different irrigation regimes. The influence of the high temperature, but not the irrigation regime, on the anthocyanin ratio is confirmed by our other experiments [25]. At the beginning of the current treatment, when air temperature was lower, the Pel3G/C3G ratio in the berries of No. 17 grown under the highest water deficit was up to 35% lower than in the berries of normally irrigated plants. 

It was established that the amount of C3G in berries of the “Rojan” variety was significantly negatively correlated with the number of berries (r = −0.83 ± 0.24) and yield of plants (r = −0.81 ± 0.26) and positively correlated (insignificantly) with the average berry weight (r = 0.56 ± 0.35) (linear correlation). On the contrary, the amount of Pel3G had a positive correlation with the number of berries (r = 0.90 ± 0.19) and a negative correlation with the average berry weight (r = 0.66 ± 0.32). Similar correlations were established in the berries of hybrid No. 17, but the correlation coefficients were lower and often insignificant.

It may be concluded that the dynamics of the yield parameters of *F. vesca* depend on the genotype and temperature changes during berry ripening. Water deficit stress causes an increase in the wild strawberry berry set and yield in the first half of harvesting and a decrease in yield parameters (with the possible exception of the yield of No. 17) at the end. Although WD caused a decrease in the total yield of up to 21.0%, the berries accumulated up to 2.3 times more anthocyanins compared with normally irrigated plants. The highest anthocyanin concentration and the lowest pelargonidin 3-O-glucoside/cyanidin-3-O-glucoside ratio (1.1–1.3) were observed in the first part of the fruiting season when the air temperature in the greenhouse was nearly 20 °C. The anthocyanin content decreased and the Pel3G/C3G ratio (3.1–3.8) increased in *F. vesc*a berries at elevated air temperatures (over 25 °C). 

The data obtained in this study expands our knowledge about growth changes during harvesting and the accumulation of anthocyanins in *F. vesca* under WD stress. These insights could help to develop cultivation technology that preserves water resources and increases the nutritional value of berries during their growth. Our data reveal that for the successful use of reduced irrigation, moderate to severe (25–50% of full irrigation) water deficit treatment at a stable air temperature of approximately 20 °C for a maximum of three weeks is desirable for berries with maximal yield and the highest nutritional value. From the obtained data, growers are recommended to reduce watering to 50% at moderate temperatures for at least the first half of the harvest season. This would reduce water consumption and improve the quality of the berries. Data on changes in anthocyanin accumulation under different environmental conditions are important for gene expression studies in order to identify key factors that affect these valuable compounds. 

## 4. Materials and Methods 

The experiment was conducted at the Lithuanian Research Centre for Agriculture and Forestry, Institute of Horticulture, Babtai Lithuania, located at the coordinates 55° 08′ N and 23° 80′ E and an elevation of 55 m. The climate is a humid continental type with an annual average precipitation of 630 mm and an average temperature of −5 °C in January and 17.3 °C in July. Wild strawberry or European “alpine strawberry” (*F. vesca* var. *semperflorens*) “Rojan” (“Rügen”), and hybrid No. 17 [(*F. vesca* (woodland strawberry)) × *F. vesca var. semperflorens*]) were used in this research. Strawberry seedlings were obtained from seeds sown in January, 2012. The seedlings were initially peeled in cartridges with peat and then planted in pots (13 × 13 × 13 cm^3^) with peat substrate in May. The properties of the peat substrate were as follows: density 0.15 (g/cm^3^), total porosity 0.78 (m^3^/m^3^), water-holding capacity 0.57 (m^3^/m^3^), pH 5.6, P_2_O_5_ 248 mg kg^−1^, and K_2_O 273 mg kg^−1^. From January to May and from November to the end of the experiment in May of the following year, the plants were grown in an unheated greenhouse. The greenhouse had a metallic frame with an arched roof, measuring 20 m long by 10 m wide, and a ceiling height of 3.5 m constructed in a north–south direction. The pots with plants were transferred and temporarily kept in open air from May to November. The resulting inflorescences were removed. Then, the pots were again transferred to an unheated greenhouse on tables with a height of 1.5 m. The pots were spaced 0.20 m apart. Plants started to bloom at the beginning of April in the following year. When grown outdoors and in the greenhouse, the plants received moisture and were fertilized according to the needs of the plants to ensure their normal growth at all times.

The water deficit treatment started on 18 April and finished on 28 May. A randomized block experimental design was used in a 3 × 2 factorial arrangement, with three water treatments and two genotypes. The plants (seedlings) were arranged in three repetitions, each consisting of 12 plants, for a total of 36 plants evaluated for each variety and irrigation treatment. During the treatment, the applied water regimes were (1) full irrigation (I100), applying 100% of the estimated water crop requirements, and (2) 50% (I50) and (3) 25% (I25), applying half and a quarter of the full irrigation water supply, respectively. Under I100, 340 mL of water per potted plant was applied. This is the average amount of water that the substrate with plant can hold without excess drainage from the pot. This volume was determined during the previous 30 days before the irrigation treatment. During the same period, it was observed that watering was required every 2–4 days to maintain sufficient substrate moisture in pots with plants. This watering frequency was maintained in the WD study, in which the plants were watered a total of 15 times. The net irrigation for fully irrigated plants during our WD treatment was 5.1 L per plant (pot). In our experience, this amount of water corresponds to the volume that is normally consumed by growing strawberry seedlings for berry production in greenhouses in early spring (data not shown). For (I50) and (I25) plants, 170 mL (50%) and 85 mL (25%) were applied, respectively, at the same periodicity as that for I100 plants. From 22 April until the end of harvesting (28 May), ripe berries were picked, counted, weighed, and frozen at −70 °C to −80 °C every fourth day. The number of berries per plant, average weight, and yield per plant were measured. The standard deviation and correlation were established using the ANOVA method and Microsoft Excel. 

The air temperature was highly variable during the experiment: the average air temperature varied from 11 °C to 24 °C, with a significant temperature increase after 8 May (Figure 4). 

The total and cumulative concentrations of anthocyanin were evaluated. The cumulative amount is the amount of anthocyanins in a particular harvest plus the amount in previous harvests. Anthocyanins were extracted from frozen berries using 90% aqueous methanol, with 0.02% HCl at a 1:20 g:mL ratio [42], and concentrated in rotary evaporator at 37 °C. Residues were dissolved in acidified water (pH 2.4). The anthocyanin composition was analyzed using an Agilent 1200 HPLC system with a DAD detector (Agilent, Germany); a reverse phase XBridge Shield RP18 (Waters, UK) analytical column (3.0 × 150 mm^2^, particle size 3.5 μm) was used for separation. The mobile phase consisted of 10% acetic acid and 1% phosphoric acid (solvent A) and 100% acetonitrile (solvent B); HPLC gradient grade reagents were purchased from Sigma-Aldrich. The elution conditions were as follows: isocratic elution 0% B, 0–12 min; linear gradient from 0% B to 7% B, 12–15 min; to 45% B, 17 min; to 100% B, 20 min, and flow rate 0.7 mL min^−1^. All anthocyanins were quantified at a detection wavelength of 520 nm using an external standard of cyanidin 3-O-glucoside (C3G), pelargonidin 3-O-glucoside (Pel3G), peonidin 3-O-glucoside (Pn3G), cyanidin 3-O-rutinoside (C3R), cyanidin 3-O-sambubioside (C3SAM), delphinidin 3-O-rutinoside (D3R), delphinidin 3-O-glucoside (D3G), peonidin 3-O-rutinoside (Peo3R), and malvidin 3-O-glucoside (M3G). All standards were purchased from Extrasynthese, France, and Polyphenols Laboratories, Norway.

## Figures and Tables

**Figure 1 plants-10-00557-f001:**
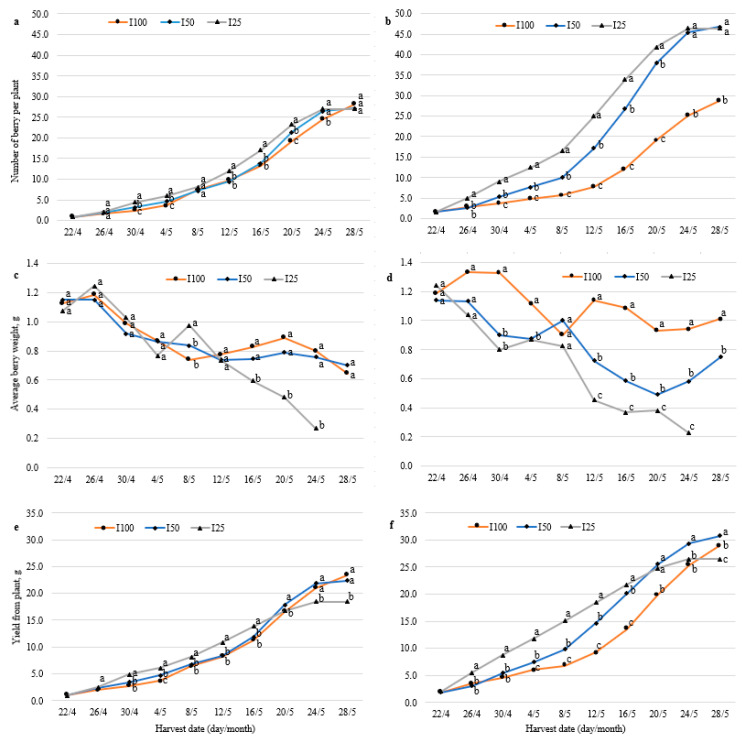
Cumulative berry number (**a**,**b**), average berry weight (**c**,**d**), and cumulative yield per plant (**e**,**f**) of alpine strawberry “Rojan” (**a**,**c**,**e**) and hybrid No. 17 (**b**,**d**,**f**) in different picks. Plants were fully irrigated (I100) and irrigated with half (I50) and a quarter (I25) of full irrigation. Data followed by the same letter are not significantly different according to the least significant difference (LSD) test at *p* ≤ 0.05.

**Figure 2 plants-10-00557-f002:**
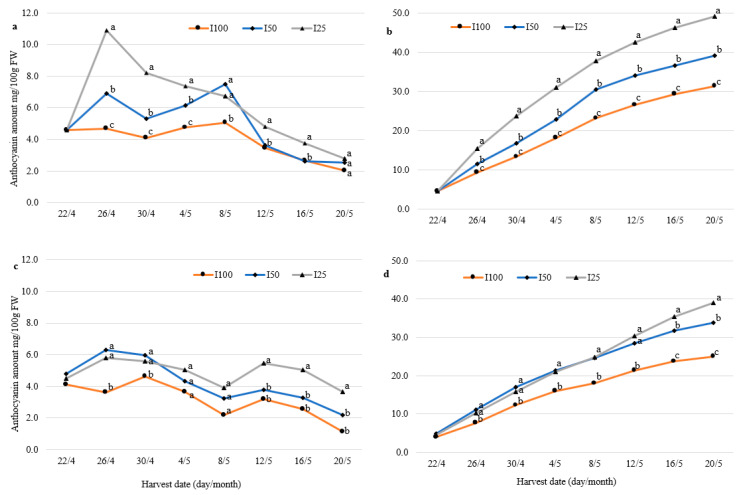
Dynamics of total anthocyanin content (mg/100g fresh weight) in the berries of wild strawberry “Rojan” (**a**) and No. 17 (**c**) and cumulative anthocyanin content (**b**,**d**), respectively. Plants were fully irrigated (I100) and irrigated with half (I50) and a quarter (I25) of full irrigation. Data followed by the same letter are not significantly different according to the LSD test at *p* ≤ 0.05.

**Figure 3 plants-10-00557-f003:**
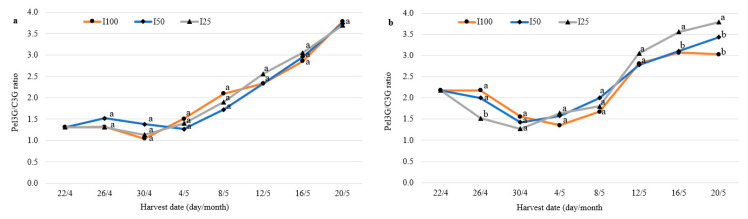
Pel3G/C3G ratio in the berries of wild strawberry “Rojan” (**a**) and No. 17 (**b**). Plants were fully irrigated (I100) and irrigated with half (I50) and a quarter (I25) of full irrigation. Data followed by the same letter are not significantly different according to the LSD test at *p* ≤ 0.05.

**Figure 4 plants-10-00557-f004:**
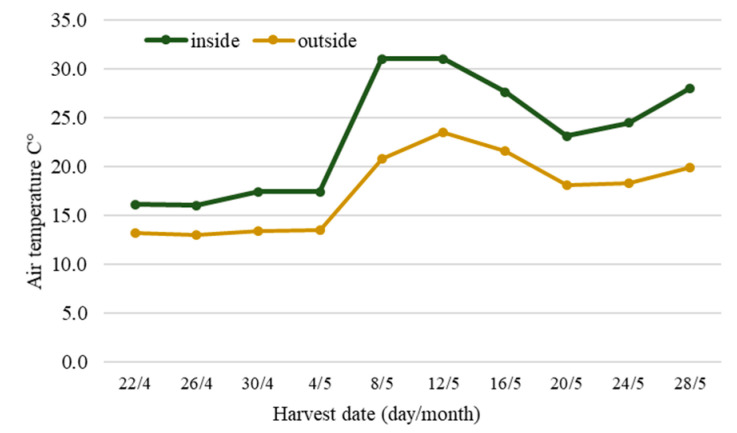
Air temperature outside and inside the greenhouse.

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
