# Peer review of "Characteristics of *Fragaria vesca* Yield Parameters and Anthocyanin Accumulation under Water Deficit Stress"

_plants, 2021, doi:10.3390/plants10030557_

Round 1
Reviewer 1 Report
The manuscript plants-1134831 had as objetive to evaluate the growth and fruiting of Fragaria vesca plants and anthocyanin content under water deficit in greenhouse. The manuscript brings interesting information about the production of this fruit under water deficit. All my comments are in the attached file. It is necessary that authors provide a good English editing before uploading the new version of the manuscript.

Author Response
Dear Reviewer,
Thank you for the detailed examination of the article and kind and valuable comments and suggestions. Corrections to the manuscript were made based on your suggestions. English editing using MDPI English Editing service was performed before resubmitting of the manuscript.
Sincerely
Authors
Reviewer 2 Report
22 February 2021
Dear authors,
it is evident that the section on irrigation in materials and methods has been significantly improved. Although, irrigation scheduling is quite confusing. It is not clear enough if the irrigation time was determined according to the plant water requirements, measurements of substrate moisture, or irrigation intervals? It is not clear enough how is crop water requirement determined? My biggest concern is that the study is not clearly presented, for example, the study before and during the harvesting. As it is stated in the manuscript, the study period is one month? What about the study period during the rest of the vegetation? Does the irrigation treatment refer only to the period of picking? Furthermore, the part where you describe the plants outside and inside the greenhouse is completely confusing!
Here are some remarques and suggestions for the material and method section:
L 309 – the study period is only one month?
L 320 – how is the plant water requirement determined?
L 320 – 322 – the growing system is unclear. First, you state that the “The experiment was carried out in the period April 18 to May 28“(L 309), then the seedling was planted in the greenhouse on May (L 317), then the „The pots with plants were transferred and kept in the open air until November (L 320), then L 321 were transferred to an unheated greenhouse
Kindly, give a clear chronological overview of how the experiment was conducted since the present method is not clear enough
L 327 – it is suggested to use 50% of ET or water requirements, not the “norm”
I encourage you to give an explanation of why did you choose 100, 50, and 25% of evapotranspiration? It is not common practice to keep plants in such drought stress!
L 329 – maybe it could be easier if you give abbreviation for irrigation treatments; i.e. I100, I50, and I25
L 328 – What is FI representing?
L 329 -331 – this is completely unclear. Is the amount of irrigation water determined according to crop water requirements by measuring the substrate moisture content?
L 333 – then you stated that the irrigation interval was 4 days!
Net irrigation for each irrigation treatment is not given!
Result section:
L82 – 83 – how do you explain the highest yield on irrigation treatment with the severe drought stress?
L 83 – Kindly, explain what did you mean by “normally irrigated”
L 87-90 This sentence is completely confusing. First, it was higher with the exception of last 4 days when it was the same and afterward… what is afterward if you have previously stated last 4 days
It is not clear enough how long did you irrigate the plants prior to the first harvest?!
Discussion
L 170 – but in your results, the yield is higher in stress conditions
L 175 – what yield parameters? Only yield data are presented
Author Response
Dear Reviewer,
Thank you for the detailed examination of the article and kind and valuable comments and suggestions. According to your comments, the methodological part about growing strawberry seedlings before and during the water deficit (WD) treatment was basically rewritten for clarity and specificity. The idea of the study was to determine how the reduction of water consumption for irrigation affects plant fertility parameters, berry quantity, size, and yield itself, as well as to determine how this affects the anthocyanin content and composition of berries. The usual amount of water used to grow strawberry plants and harvest in an unheated greenhouse in the spring is known and calculated to us, because wild strawberry were grown here for several years in a greenhouse in aim to obtain early yield. In order for the plants to grow and strengthen for next year’s harvest, they were grown outdoors during the season, part of time, as this facilitated their care and protection from pests, which is difficult in a greenhouse during the whole year. In Lithuanian conditions, in order for the plants to be strong enough for a high-quality and maximum yield, usually they are planted year before harvest. The resulting first year inflorescences were removed. The plants were watered as needed so that their growth would be fully satisfied for it, and the substrate surface would remain moist but there would be no excess water. This amount was also estimated in the study year, before WD treatment. During the WD experiment, which began with the onset of fruit set, watering levels for control plants were maintained and for WD plants reduced by half or three quarters. We are convinced that this reduction is significant enough. The influence of WD on the above parameters was evaluated. All control and 50% irrigated plants survived to the end of cropping. Plants irrigated 25% wilted and died before the end of the cropping.
Answers to the Reviewers comments and suggestions:
L 309 (now L334) – the study period is only one month?
Answer: Plants growth was more than one year, but WD experiment lasts only cropping time – last five weeks.
L 320 (now 331) – how is the plant water requirement determined?
Answer: When grown outdoors and in the greenhouse, the plants received moisture and were fertilized according to the needs of the plants to ensure their normal growth. Water was added to the pots with the substrate and the plants so that no excess of it would flow out of the pot. The watering were carried out at such a frequency that the substrate could not dry out, it always remained moist. On hot days it was watered on average every two days, on cool cloudy days every four days. Such amount of watering for all plants and in WD treatment for the control plants was maintained at all the time.
L 320 – 322 (now 324-334) – the growing system is unclear. First, you state that the “The experiment was carried out in the period April 18 to May 28“(L 309), then the seedling was planted in the greenhouse on May (L 317), then the „The pots with plants were transferred and kept in the open air until November (L 320), then L 321 were transferred to an unheated greenhouse
Kindly, give a clear chronological overview of how the experiment was conducted since the present method is not clear enough
Answer: the cultivation system was such as to ensure maximum yield in the spring of the following year. This is described in the revised methodological section of the manuscript. In short: the seeds were sown in January, the seedlings peeled and planted in pots and grown in the greenhouse until May. From May to November, the plants in pots were kept and cared outdoors. In November, the pots with the plants were moved back to the unheated greenhouse on the tables for overwintering and accelerating spring harvest. Further, until the end of watering experiment in May (which started in April at the beginning of harvesting), the plants were grown in the same greenhouse.
L 327 (now 338) – it is suggested to use 50% of ET or water requirements, not the “norm”
Answer: We totally agree. We have changed this in the text by no longer using the term "norm"
I encourage you to give an explanation of why did you choose 100, 50, and 25% of evapotranspiration? It is not common practice to keep plants in such drought stress!
Answer: We have chosen such watering amount to cause a water deficit because we believe that such a reduction will be significant enough, knowing that a small water deficit is well tolerated by plants with little or no consequences.
L 329 (now 338) – maybe it could be easier if you give abbreviation for irrigation treatments; i.e. I100, I50, and I25
Answer: We agree and gave abbreviation for irrigation treatments I100, I50, I25.
L 328 – What is FI representing?
Answer: FI means full irrigation. We removed this term in revised manuscript text.
L 329 -331 – (now 341-350) this is completely unclear. Is the amount of irrigation water determined according to crop water requirements by measuring the substrate moisture content?
Answer: The substrate in pots with plants under full irrigation was watered at such a frequency that substrate visually remained constantly moist. According our previous measurements before next watering substrate reaches 60% water-holding capacity this is about 0.34 (m3/m3). As mentioned, the control plants and all plants before WD treatments, was watered in such a regime and in such quantities as to ensure optimal plant growth, without loss of drained water.
L 333 – (now 345) then you stated that the irrigation interval was 4 days!
Answer: There were a mistake in the manuscript. In fact, the frequency of watering was 2-4 days, depending on the air temperature inside the greenhouse and the rate of evaporation. This section has been corrected.
Net irrigation for each irrigation treatment is not given!
Answer: Net irrigation for control plant during the WD treatment was 5.1 l per plants. For I50 and I25 there was 2.6 and 1.3 l per plant respectively.
Result section:
L82 – 83 – how do you explain the highest yield on irrigation treatment with the severe drought stress?
Answer: This highest yield in treatment with severe drought stress was only temporary. WD cause berry ripening acceleration. WD stress, especially for I25 plants resulted in an increase in berry number, but a decrease in size from the middle of the study and a decrease in yield in the last weeks of the study.
L 83 – Kindly, explain what did you mean by “normally irrigated”
Answer: “Normally” means irrigation as usually, or full irrigation, meeting the needs of plants by growing plants for normal berry production.
L 87-90 (now 87-89) This sentence is completely confusing. First, it was higher with the exception of last 4 days when it was the same and afterward… what is afterward if you have previously stated last 4 days
Answer: This sentence was modified: “The cumulative yield of I25 plants was higher than that of I100 plants until the 16th of May, which is about two-thirds of the duration of the treatment. During the last 4 days, it became the lowest compared with I50 and control plants.”
It is not clear enough how long did you irrigate the plants prior to the first harvest?!
Answer: up to 4 days before the first harvest all the plants were irrigated according plant requirements (as explained before).
Discussion
L 170 (now 177) – but in your results, the yield is higher in stress conditions
Answer: Deep WD when 25% of full irrigation was applied decreased final yield in our experiment. At the beginning, when WD accelerated berry ripening, berry yield increased. The conclusion is that in order to improve anthocyanin synthesis and at least not reduce yields, a small or moderate WD can be applied for a limited period.
L 175 (now 182) – what yield parameters? Only yield data are presented
Answer: we assume that the yield parameters are the number of berries and the size of the berries.
English editing using MDPI English Editing service was performed before resubmitting of the manuscript.
Sincerely
Authors
Reviewer 3 Report
Dear Authors,
the manuscript ‘Characteristic of Fragaria vesca yield parameters and anthocyanin accumulation under water deficit stress’ possesses good significance of the content, for the application in horticultural research. However, the originality/novelty and the quality of the presentation (including English) are low.
In summary, general comments:
- The English needs to be revised. Some parts must be re-phrased for better understanding.
- The journal’s template has not been followed strictly (for sub-paragraphs heading, and for Reference list). The List of reference in particular… why the title of the article is with capital letters? Pls check with the Editor and journal template.
Specific comments are reported in the attached file.
I recommend the authors to answer thoroughly point by point to the reviewer’s comments, highlighting where the changes have been done. This is mandatory in order for the manuscript to be accepted.

Author Response
Dear Reviewer,
Thank you for the detailed examination of the article and kind and valuable comments and suggestions. Corrections to the manuscript were made based on your suggestions. English editing using MDPI English Editing service was performed before resubmitting of the manuscript.
Answers to specific comments:
Line 124 (now 127) “which variety? This phrase is obscure. Are you referring to your results or it is a statment from bibliography? Pls, clarify and re-phrase”.
Answer: We re-phrase: ”It was observed in our study that WD caused considerable increase of anthocyanin amount in wild strawberry fruits especially at the beginning of the harvest.” It was observed in our study that WD caused a considerable increase in anthocyanin content in wild strawberry fruits, especially at the beginning of the harvest.”
It is valid for both investigated varieties, but this was more pronounced for variety No 17. This result we obtained in our study.
Line 178 (now 185) “It is reported that ...(and cite ref for this as it is a well know phenomenon due to adaptation of plant to stress)”
Answer: The citation is presented above. We re-phrase the sentences with citations: “WD stress has been reported to stimulate both berry set and ripening speed [23]. Mild WD was found to increase apple yield [24].”
Line 196 (now 202) “It is not clear here if you are dealing with your results in the experiments or the mentioned 'three stages' is a general feature, then cite the ref. Pls, re-phrase!”
Answer: We re-phrase the sentence: “In our opinion, the changes in yield parameters under water deficit conditions in our treatment can be divided into three stages. In the first stage…”
The division into three phases are only our considerations when summarizing our results. We did not find such a division in the literature.
Line 246 (now 253) “as it has been observed in other studies (36; Azuma A, Yakushiji H, Koshita Y, Kobayashi S. Flavonoid biosynthesis-related genes in grape skin are differentially regulated by
temperature and light conditions. Planta. 2012;236:1067-80.; Blando et al, (2018) J. Berry Res. DOI: 10.3233/JBR-170258)”
Answer: we very grateful for your kind suggestion to add cited sources. We were happy to add them in our manuscript.
Line 336 (now 354)“re-phrase, not clear”
Answer: “Standard deviation and correlation were established using the ANOVA method and Microsoft Excel”.
Corrections were made also in Reference list and sub-paragraphs heading.
Sincerely
Authors
Reviewer 4 Report
Dear authors,
Congratulations for this nice work. I only found one small format error in line 332, please remove the letter "l" in "mLl".
Best regards,
Author Response
Dear Reviewer,
Thank you for the detailed examination of the article and positive feedback! Corrections to the manuscript were made based on your suggestions. English editing using MDPI English Editing service was performed before resubmitting of the manuscript.
Sincerely
Authors
Round 2
Reviewer 2 Report
Dear Authors,
Dear authors, thank you for responding to the comments andexplaining what was not quite clear in the paper.
I think that after the corrections were made, the work gained in quality
Best luck in future scientific work
This manuscript is a resubmission of an earlier submission. The following is a list of the peer review reports and author responses from that submission.
Round 1
Reviewer 1 Report
The present research deals with an important issued faced by the world nowadays which is the water scarcity. In this sense research on the production of hydroSOStainable products is of utmost importance. For this reason, I consider this article for publication in Plants journal, after the correction of the following aspects:
ABSTRACT
Line 16: Please remove “stress”
Line 17: Does “of norm” means “control”, please explain.
Line 20: Here the authors explains that the yield depended on the genotype if many cultivars were evaluated please specify in Line 16 which ones. Moreover, does “cropping” means “harvesting”?
Line 21: Please change the word “however” with “additionally” or “in addition” to highlight that beside yield the product is higher in important bioactive compounds raised by the water stress.
Line 23: Change “normally” by “full irrigated”.
Line 24: “under stabile close” is not understandable, please rewrite this sentence.
Line 25: correct as “positively affected the secondary plant metabolites such as anthocyanins and also the yield of….”
INTRODUCTION
Line 29: Correct as “word”
Line 35-36: Correct as “…and the concentration of health-related compounds and taste in strawberry fruits”
Line 57: Correct as “is relative to cultivated”
Line 60: Correct as “under water deficit in greenhouse”
Line 60-61: Please remove the last sentence, in the introduction section must end with the purpose, this sentence is a conclusion which might me moved to the abstract, conclusion section or deleted.
MATERIALS AND METHODS
Line 280: Please add “in this research”
Line 281: Correct as “The experiment was carried out from April 18 to May 28”. Because, above authors already mentions that harvest time it started from 22 April to 28 May. Authors must make sure that the correct term of harvesting for berries is “cropping”, if yes keep it, if not change it by harvest, so it is easier to understand.
Line 284: Correct as “according to its…”
Line 286: Correct as “mL” here and through the manuscript.
Line 290: Correct as “were used for analysis”.
Line 291: Anova is a statistical method, not a software please add the used software.
Line 292: Correct as “during experiment”
RESULTS
Lines 68: Correct as “by extreme air temperature which increased up to…at the beginning of 8th of May.”
Line 112: “Anthocyanin profile and content in berries during …”
Line 115: It was observed that water deficit….
Line 100: I would like to see marked in all graphics the significant differences between the treatments. For instance, in Figure 1 e, in the day 28/5 I would like to see marked by Tukeys letters the differences among treatments. I see the 25 % very low, that one I suppose is a c and represents the lowest yield, but if is not written we don’t know. However, between 50 % and 100 % the difference is less but I don’t know if is significant or not. At least in the most important harvesting days.
Line 129: What is the difference between total anthocyanins and cumulative? Please explain through the manuscript to be clear.
Lines 134-140: I recommend a Table or a Graphic chart which will clearly show these results.
Line 177: “decreased at the end of”
Line 260-261: is not totally true, if we check the figures about yield, we can see that in the Genotype N17, the yield was not reduced by water stress at the end of the harvesting. Please rewrite.
CONCLUSION
I miss in this part and also in the conclusion of the abstract a sentence in which the authors recommend the best irrigation treatment in terms of yield and anthocyanins balance, and how much irrigation water is saved using this treatment.

Reviewer 2 Report
Dear Authors,
the manuscript ‘Characteristic of Fragaria vesca yield parameters and anthocyanin accumulation under water deficit stress’ deals with the characterization of agronomic parameters (number of fruits, average berry weight, yield) as well bioactive (anthocyanins) compounds content.
The main impression I had from this manuscript is not positive, since the are many flaws in my opinion, which I try to summarize afterward. Moreover, although there is good significance of the content, the originality/novelty, the quality of presentation (including English) and scientific soundness are low.
In summary, some week points are the following:
- Along the manuscript, including the abstract, there are some parts taken from a previous paper from the authors (ref. 27) (self-plajarism).
- In my opinion, the figures are not very accurate: no SD is shown along the different points of the graph, and figure captions are not accurate (see comments directly on the file).
- The English need to be revised.
- The presentation of Results and the Discussion of results is not linear, not easy to follow. The authors should try to reformulate.
- The Methodology needs to be amended for anthocyanin standards.
- The List of references is too long. The authors should need to cite the more relevant papers for the discussion of their topic. Moreover, they should write the references in the correct way following journal instruction.
Reviewer 3 Report
Dear authors, please find my comments in the word document

Reviewer 4 Report
Dear Editor, all my comments and suggestions are in the attached file. The main problem detected in the manuscript is the lack of statistical analysis. Authors only presented line graphics (with means over time), but no conclusion can be done without the ANOVA test. In this trial there are 3 water deficit treatments and 2 strawberry genotypes, but no statistical comparison was presented. Authors must present the statistical analysis in order to support the statements. Results and Discussion section must be redone according to these changes.
